Gene expression of benthic amphipods (genus: Diporeia) in relation to a circular ssDNA virus across two Laurentian Great Lakes

http://orcid.org/0000-0002-1664-5028 Bistolas Kalia S.I. 1 ksb97@cornell.edu
Rudstam Lars G. 2
Hewson Ian 1
1 Department of Microbiology, Cornell University , Ithaca, NY , USA
2 Department of Natural Resources and the Cornell Biological Field Station, Cornell University , Bridgeport, NY , USA
Prentis Peter
Electronic publication date: 2017 Sep 26
Publication date: 2017
Volume: 5
Electronic Location ID: e3810
Received 2017 Jun 29; Accepted 2017 Aug 23
Copyright: © 2017 Bistolas et al.
Copyright year: 2017
Copyright holder: Bistolas et al.
License: This is an open access article distributed under the terms of the Creative Commons Attribution License, which permits unrestricted use, distribution, reproduction and adaptation in any medium and for any purpose provided that it is properly attributed. For attribution, the original author(s), title, publication source (PeerJ) and either DOI or URL of the article must be cited.
License URL: https://creativecommons.org/licenses/by/4.0/

Keywords: Diporeia, CRESS-DNA, Laurentian Great Lakes, Transcriptomics, ssDNA virus

Funding: National Science Foundation NSF-135696 and DGE-1144153 U.S. Environmental Protection Agency, Cooperative Agreement GL 00E01184-0 This research is supported by the National Science Foundation (NSF-135696 and DGE-1144153) with additional support from the U.S. Environmental Protection Agency, Cooperative Agreement GL 00E01184-0 to Cornell University. The funders had no role in study design, data collection and analysis, decision to publish, or preparation of the manuscript.

==============================
Circular rep-encoding ssDNA (CRESS-DNA) viruses are common constituents of invertebrate viral consortia. Despite their ubiquity and sequence diversity, the effects of CRESS-DNA viruses on invertebrate biology and ecology remain largely unknown. This study assessed the relationship between the transcriptional profile of benthic amphipods of genus Diporeia and the presence of the CRESS-DNA virus, LM29173, in the Laurentian Great Lakes to provide potential insight into the influence of these viruses on invertebrate gene expression. Twelve transcriptomes derived from Diporeia were compared, representing organisms from two amphipod haplotype clades (Great Lakes Michigan and Superior, defined by COI barcode sequencing) with varying viral loads (up to 3 × 106 genome copies organism−1). Read recruitment to de novo assembled transcripts revealed 2,208 significantly over or underexpressed contigs in transcriptomes with above average LM29173 load. Of these contigs, 31.5% were assigned a putative function. The greatest proportion of annotated, differentially expressed transcripts were associated with functions including: (1) replication, recombination, and repair, (2) cell structure/biogenesis, and (3) post-translational modification, protein turnover, and chaperones. Contigs putatively associated with innate immunity displayed no consistent pattern of expression, though several transcripts were significantly overexpressed in amphipods with high viral load. Quantitation (RT-qPCR) of target transcripts, non-muscular myosin heavy chain, β-actin, and ubiquitin-conjugating enzyme E2, corroborated transcriptome analysis and indicated that Lake Michigan and Lake Superior amphipods with high LM29173 load exhibit lake-specific trends in gene expression. While this investigation provides the first comparative survey of the transcriptional profile of invertebrates of variable CRESS-DNA viral load, additional inquiry is required to define the scope of host-specific responses to potential infection.

Introduction

Circular rep-encoding ssDNA (CRESS-DNA) virus genomes are small (∼1.7–4 kb), circular molecules which encode, at minimum, major open reading frames rep (replication initiator protein) and cap (structural capsid protein; Rosario, Duffy & Breitbart, 2012; Rosario et al., 2015). Eukaryotic CRESS-DNA viruses broadly encompass ssDNA viruses that infect plants (Geminiviridae, Nanoviridae), and metazoans (Circoviridae, Anelloviridae; Dunlap et al., 2013; Rosario et al., 2017; Rosario, Duffy & Breitbart, 2012), and include common and important pathogens of ecologically or commercially relevant vertebrates. For example, beak and feather disease virus (BFDV, Circoviridae) is responsible for persistent immunosuppression in avian hosts (Eastwood et al., 2014) and porcine circoviruses infect domestic swine, manifesting sub-clinically (PCV1) or eliciting postweaning multisystemic wasting syndrome (PMWS, PCV2; Allan & Ellis, 2000). The use of culture-independent (metaviromic) approaches has led to the discovery and characterization of an extraordinary diversity of novel ssDNA viruses in environmental reservoirs and non-model invertebrates (Labonté & Suttle, 2013; Rosario & Breitbart, 2011; Rosario et al., 2017; Rosario, Duffy & Breitbart, 2012; Roux et al., 2016). To date, the etiology, pathology, and association between ssDNA viruses and any invertebrate remains wholly unknown. This study utilized whole transcriptome sequencing to investigate the relationship between a CRESS-DNA virus and benthic amphipods of genus Diporeia from the Laurentian Great Lakes.

Circular rep-encoding ssDNA viruses have been identified in association several major aquatic invertebrate phyla, including the Annelida, Arthropoda, Chaetognatha, Cnidaria, Ctenophora, Echinodermata, and Mollusca, among others (Breitbart et al., 2015; Dayaram et al., 2016; Dunlap et al., 2013; Eaglesham & Hewson, 2013; Fahsbender et al., 2015; Kibenge & Godoy, 2016; Hewson et al., 2013a, 2013b; Jackson et al., 2016; Rosario et al., 2015; Soffer et al., 2013). These viruses appear to be biogeographically widespread, taxonomically diverse, and common constituents of crustacean nanobiomes (Dunlap et al., 2013; Hewson et al., 2013a, 2013b; Labonté & Suttle, 2013; Rosario et al., 2015, 2017; Rosario, Duffy & Breitbart, 2012). However, little is known about the role of CRESS-DNA viruses in mediating crustacean ecology, physiology, and mortality. Because no immortal crustacean cell line currently exists, propagation of crustacean-associated CRESS-DNA viruses in vitro remains intractable, and the unknown nature of CRESS-DNA virus tropism and infection dynamics in these systems impedes targeted sequencing of virus-infected cells. Furthermore, many microcrustaceans cannot be reared or maintained effectively in aquaria without significant physiological stress and high incidence of mortality, hindering in vivo infection experiments. Therefore, we implemented a whole-organism comparative transcriptome sequencing (transcriptomics) approach in evaluating the relationship between the presence of CRESS-DNA viral genotype, LM29173, and benthic crustaceans (genus: Diporeia) in Great Lakes ecosystems.

Diporeia are historically abundant benthic meiofauna in the Laurentian Great Lakes (Auer et al., 2013; Barbiero et al., 2011; Birkett, Lozano & Rudstam, 2015; Guiguer & Barton, 2002). These amphipods influence lake-wide biogeochemistry and mediate relationships between spring diatom blooms and upper trophic level consumers through detritivory and sediment bioturbation (Gardner et al., 1985; Guiguer & Barton, 2002; Halfon, Schito & Ulanowicz, 1996; Wells, 1980). Localized and precipitous declines in several Diporeia populations have prompted exploration of their viral consortia (Bistolas et al., 2017; Hewson et al., 2013a). Metavirome sequencing has documented a common and recurrent CRESS-DNA virus genotype, LM29173, frequently detected in impacted Diporeia populations in Lakes Michigan and Huron, but rare among specimens from stable Lake Superior populations (Bistolas et al., 2017; Hewson et al., 2013a). It is also prevalent among amphipods from the deep, glacial Finger Lakes of Central New York (Seneca, Cayuga, and Owasco Lakes). Previous DNA barcoding of maternally inherited cytochrome c oxidase I (COI) sequences (Pilgrim et al., 2009; Bistolas et al., 2017) have revealed sub-species genetic variation between impacted and stable populations, with Diporeia from Lakes Michigan, Huron, Ontario, Erie, and the Finger Lakes comprising a southern lake haplotype clade, and amphipods from Lake Superior comprising a northern lake haplotype clade. While LM29173 is more abundant in Diporeia from declining southern populations than stable northern populations, no advances have been made to describe the impact of this CRESS-DNA virus on amphipod biology. This study offers preliminary insight into the relationship between LM29173 and gene expression in amphipods from both haplotype clades, and provides transcriptional targets for further investigation. Specific objectives of this study were to (1) investigate the association between LM29173 presence/load and the transcriptional profile of Diporeia, (2) determine if detected changes in gene expression are specific to distinct Diporeia haplotypes, and (3) explore the effect of LM29173 presence on amphipod transcription of innate immunity regulators/effectors.

Materials and Methods

Sample collection and transcriptome preparation

Diporeia were collected in August–September 2014 via Ponar benthic sampler from Great Lakes Michigan and Superior at EPA-designated stations (Fig. 1, Table S1; United States Environmental Protection Agency, 2012). Organisms were sieved to remove sediment (500 μm), rinsed, and immediately individually frozen at −80 °C.

Figure 1 Amphipod collection sites in the Laurentian Great Lakes (August–September, 2014).

Collection locations are congruent with EPA-Great Lakes National Program Office (GLNPO) designated stations (United States Environmental Protection Agency, 2012). Specimens were collected on the R/V Lake Guardian via Ponar benthic sampler. Bathymetry data was provided by NOAA National Geophysical Data Center’s Marine Geology & Geophysics Division (NGDC/MGG) and the NOAA Great Lakes Environmental Research Laboratory (GLERL). Map service published and hosted by Esri Canada© 2012 under Attribution-NonCommercial 2.5 Canada (CC BY-NC 2.5 CA) license https://creativecommons.org/licenses/by-nc/2.5/ca/.

Nucleic acids were extracted from individual amphipods via ZR-Duet™ DNA/RNA MiniPrep kit (Zymo Research, Irvine, CA, USA). Presence and genome load (copy number) of LM29173 was determined via qPCR per Hewson et al. (2013a) using SsoAdvanced™ Universal Probes Supermix (Bio-Rad Laboratories, Hercules, CA, USA), corrected for total extraction volume, and standardized by organism wet weight (mg). Two samples with the highest and two samples with the lowest copy numbers organsim−1 of LM29173 from each of three stations (Lake Michigan 27 and 40, Lake Superior 066; Fig. 2, Fig. S1; United States Environmental Protection Agency, 2012) were selected for transcriptome preparation (n = 12 total transcriptomes, n = 4 per station). For selected samples, RNA fractions were further enzymatically digested with TurboDNAse (Thermo Fisher Scientific, Waltham, MA, USA) for 15 min to reduce co-extracted DNA. Ribosomal RNA was depleted via mRNA-ONLY™ mRNA Isolation Kit (Epicentre, Madison, WI, USA), and remaining RNA was reverse transcribed and amplified via the TransPlex® Complete Whole Transcriptome Amplification Kit (WTA2; Sigma-Aldrich, Saint Louis, MO, USA) per manufacturer instructions. Resulting cDNA libraries were quantified via PicoGreen fluorescence and prepared for sequencing using a Nextera XT DNA Library Preparation Kit (Illumina, San Diego, CA, USA). Resulting libraries were subjected to 2 × 250 bp paired-end sequencing on an Illumina MiSeq at the Cornell University Core Laboratories Center (Ithaca, NY, USA). Libraries were deposited in Genbank (accession: PRJNA379017; SRR5341776–SRR5341788).

Figure 2 Quantitative detection of LM29173.

(A) Prevalence and average load (log10 transformed copy number mg−1 of tissue ± 1SE) of CRESS-DNA virus genotype LM29173 in amphipods from Great Lakes Michigan, Huron, and Superior. Viral load was significantly greater in Lake Michigan than Lakes Huron (Games–Howell post hoc t = 7.30, p = 3.1 × 10−9) or Superior (Games–Howell post hoc t = 7.30, p = 3.0 × 10−9); (B) Load of LM29173 (copy number organism−1) in amphipods selected for transcriptome sequencing. Four samples (two with the highest and two with the lowest viral load) were selected from each of three stations: Lake Michigan 27 (Mi27), Lake Michigan 40 (Mi40), and Lake Superior 066 (Su066). Individual transcriptomes are denoted by sample ID (#71, 72, 75, 77, 121, 128, 139, 130, 358, 359, 361, and 362).

Transcriptome assembly and comparison of transcript expression

Reads were trimmed for quality (quality score < 0.05, modified-Mott trimming algorithm), ambiguous nucleotides (n = 0), length (50 nt ≤ length ≤ 251 nt), and Illumina adapters via CLC Genomics Workbench (v.8.5.1; Qiagen, Hilden, Germany). Reads mapped to SILVA rRNA databases (90% identity, 50% coverage via CLC Genomics Workbench; http://www.arb-silva.de/) were excluded from assembly. Remaining reads were then assembled de novo using Trinity on the Galaxy bioinformatics platform per default parameters (National Center for Genome Analysis Support, Indiana University Pervasive Technology Institute; Table S2). Resulting contigs were further clustered via CD-HIT-EST to reduce isoform redundancy (sequence identity cutoff = 0.98). Reads were aligned to contigs via the Bioconductor package EdgeR (Robinson & Smyth, 2007) in CLC Genomics Workbench (v.8.5.1; Qiagen, Hilden, Germany) to calculate relative read recruitment (reads per kilobase of transcript per million mapped reads; RPKM) and significance (corrected for multiple comparison via false discovery rate methods; FDR). Contigs that exhibited >10-fold change (EdgeR) in read recruitment, ΔRPKM > 100, and FDR-adjusted p < 0.05 between the six low LM29173 load libraries and six high LM29173 load libraries were considered significantly differentially expressed genes (DEGs). DEGs were then annotated using Blast2Go (v.4.0.7 BLASTx, e < 1 × 10−5) and functionally classified by EuKaryotic Orthologous Group, or “KOG” (Joint Genome Institute).

Eight Lake Michigan libraries were grouped by station and viral load to identify DEGs common between both stations in Lake Michigan, minimizing the effect of between-lake genetic and environmental variance. DEGs shared between libraries were defined per the following criteria: >2-fold change in expression, ΔRPKM > 10 between libraries, significantly differentially expressed with an FDR-adjusted p < 0.05, and consistently over or underexpressed in both Lake Michigan stations. Contigs fulfilling these criteria were annotated via BLASTx against the non-redundant (nr) database and assessed for relevance to viral infection (Altschul et al., 1990).

To identify contigs affiliated with putative immune functions, reference sequences associated with invertebrate innate immunity were collected from the Insect Innate Immunity Database (Brucker et al., 2012) or curated from NCBI protein database queries of keywords in Table 1 of McTaggart et al. (2009) (keywords listed in Table S3). Contigs homologous to these genes were identified via BLASTx (e < 1 × 10−5; Altschul et al., 1990), and the RPKM of those that were >2-fold over or underexpressed in both Lake Michigan stations (Mi27 and Mi40) were standardized to total 18 s rRNA RPKM per library and depicted via web-based visualization tool, Morpheous (Broad Institute, Cambridge, MA, USA).

Table 1 Total number of over- and underexpressed contigs in transcriptomes with above average LM29173 load.

Library	Overexpressed	Underexpressed	Total	
SU066	169	65	234	
MI40	129	161	290	
MI27	1,497	187	1,684	
Notes:

Contigs that exhibited >10-fold change (EdgeR), ΔRPKM > 100, and FDR-adjusted p < 0.05 were considered significantly differentially expressed genes (DEGs).

Quantification (RT-qPCR) of differentially expressed target genes

Whole amphipods were collected in August–September, 2014 at EPA-designated stations (United States Environmental Protection Agency, 2012, Table S1) in Lakes Michigan, Huron, and Superior and extracted via ZR-Duet™ DNA/RNA MiniPrep kit (Zymo Research, Irvine, CA, USA). Load of LM29173 was quantified per Hewson et al. (2013a). RNA was reverse transcribed (RT) via Superscript III (Invitrogen, Carlsbad, CA, USA per manufacturer instructions). Parallel no-RT controls were generated using identical reaction parameters and no reverse transcriptase. cDNA was subjected to duplex RT-qPCR (quantifying both a gene of interest and a reference gene to control for organism variability) using SsoAdvanced™ Universal Probes Supermix (Bio-Rad Laboratories, Hercules, CA, USA). Amplicons were gel-purified (Zymoclean™ Gel DNA Recovery Kit; Zymo Research, Irvine, CA, USA) and cloned (pGEM®-T Easy Vector; Promega, Madison, WI, USA) using JM109 competent E. coli (Invitrogen, Carlsbad, CA, USA). Plasmids were extracted per Zyppy™ Plasmid Miniprep Kit instructions (Zymo Research, Irvine, CA, USA) and Sanger sequenced (Cornell University Core Laboratories Center, Ithaca, NY, USA) to confirm primer/probe specificity. Reaction parameters and primer, probe, and standard sequences are detailed in Table S4.

Samples were run in duplicate with congruent duplicate no-RT controls and quantified using duplicate eight-fold standard dilutions (limits of detection described in Table S4). Ct values, quantity, and standard deviation between technical replicates were determined via StepOnePlus software v.2.3 (Foster City, CA, USA). Valid runs were defined by reaction efficiency >94% and standard regression linearity (R2) > 0.98. Samples were excluded if Ct standard deviation between replicates was >0.5. Quantities were corrected for total extraction and reverse transcription dilutions. Quantities of targets β-actin (ACT), ubiquitin-conjugating enzyme E2 (UBQ), and non-muscular myosin heavy chain (NMHC) were standardized by copy number of elongation factor-1α (EF1A) per reaction.

Results and Discussion

Investigation of amphipod transcriptomes revealed differential expression of DNA replication/repair pathways, cytoskeletal architecture, and post-translational modification associated genes in correlation with CRESS-DNA virus load. However, the degree of variability between transcriptomes limited the ability to identify over or underexpression of specific molecular pathways. It is unknown whether vertebrate and invertebrate CRESS-DNA viruses utilize similar pathways of infection, particularly in light of the considerable divergence in sequence homology and genome architecture between groups. Despite this, DEGs in Diporeia transcriptomes were often homologous to DEGs in porcine circoviral infections, or were associated with putative innate immune functions. Expression of these genes varied between amphipod haplotype clades, suggesting that the transcriptional relationship with LM29173 may have a heritable component. It remains unclear if CRESS-DNA viral load corresponds significantly with ecologically relevant changes in invertebrate physiology.

Detection of LM29173

Prevalence and load of LM29173 was significantly greater in Lake Michigan (100%) than Lakes Huron (66.7%; Games–Howell, p < 1 × 10−8, Ruxton & Beauchamp, 2008) and Superior (30.8%; Games–Howell, p < 1 × 10−8; Ruxton & Beauchamp, 2008; Welch’s ANOVA, F2,26.24 = 26.4, p = 5.21 × 10−7), congruent with previous observations of the distribution of this genotype (Bistolas et al., 2017). Pilgrim et al. (2009) utilized mitochondrial COI sequences to identify sub-species genetic variation between Diporeia populations among Great Lakes ecosystems, ultimately delineating two clades with distinct haplotype signatures. qPCR results corroborate previous observations that LM29173 is detected in greater abundance in southern lakes haplotype clade populations (Lakes Michigan, Huron, Ontario, Erie, and the Finger Lakes), relative to northern lakes haplotype clade populations (Lake Superior; Bistolas et al., 2017). Because LM29173 was positively detected in all Lake Michigan amphipods, samples with the highest and lowest respective load of LM29173 were utilized for transcriptome preparation (Fig. 2; Fig. S1).

Transcriptome assembly and annotation of DEGs

Sequence reads from twelve Diporeia transcriptomes were collated (n = 14,702,859 after trimming and exclusion of rRNA-like sequences) to de novo assemble 82,074 contigs with a mean length of 310 nt and N50 value of 290 nt (Fig. S2; Tables S2 and S5). Despite rRNA depletion prior to sequencing, computational subtraction of rRNA reads was considerable (0.016–36.95%), but comparable to previously observed proportions in other studies (Schmieder, Lim & Edwards, 2012; Stewart, Ottesen & DeLong, 2010). Less than 1.25% of all rRNA-mapped reads (90% identity, 50% coverage; SILVA rRNA database) were putatively bacterial in origin, indicating that co-infecting microbes may contribute to variation in Diporeia transcriptional profiles. No transcripts of non-target CRESS-DNA viruses were identified. However, 19 unique contigs (376–6,914 nt) shared sequence similarity to putative metazoan-associated RNA viruses when compared to a manually curated database of viral RNA-dependent RNA polymerase sequences (GenBank) or the non-redundant database (BLASTx; e < 1 × 10−5). These contigs were homologous to members of the Nodaviridae (n = 4), Nyamiviridae (n = 1), Orthomyxoviridae (n = 1), Peribunyaviridae (n = 6), Phenuiviridae (n = 2), and Rhabdoviridae (n = 5), yet it remains unclear if these sequences represent transient/nonpathogenic viruses or specific pathogens of Diporeia. Furthermore, despite methodological biases favoring amplification of encapsidated RNA viruses, read recruitment to these contigs was negligible (5.74 × 10−05—0.24% of mapped, non-rRNA reads), indicating that these genotypes may have minimal relative impact on overall amphipod transcription.

Due to the lack of a reference Diporeia genome, transcripts were conservatively assembled using an isoform-sensitive algorithm, resulting in a fragmented assembly with multiple isoforms per gene. To reduce redundant read mapping, contigs were grouped into 59,317 isoform clusters prior to recruitment analysis. Library D130 (Lake Michigan site Mi40; Table S5) contained fewer total reads relative to other libraries, but was retained, as relative read recruitment was standardized by sequencing depth per library. The statistical package, EdgeR, detected 2,208 significantly DEGs between libraries with high and low LM29173 load among three Great Lakes stations (Table 1). Correlative multidimensional scaling (MDS) analyses indicated that transcriptomes do not cluster by viral presence, viral load, station, or haplotype, likely as a result of high variability in ontogeny and life history between organisms (Fig. S3). Libraries from Mi27 (Lake Michigan) contained over seven-fold more DEGs than Mi40 (Lake Michigan) or Su066 (Lake Superior) libraries. 89% of these transcripts were overexpressed in libraries with high LM29173 load (Table 1; Fig. 3) but were small, unannotated, contained no ORFs, and were therefore removed from downstream analyses. Conversely, volcano plots (Fig. 3) illustrate a roughly symmetrical distribution of significantly over and underexpressed genes in libraries from Mi40 and Su066 relative to viral load.

Figure 3 Volcano plots depicting the distribution of differentially expressed contigs.

Distribution of differentially expressed contigs was determined by EdgeR (Robinson & Smyth, 2007) for libraries from each of three stations: Superior 066 (Su066), Michigan 40 (Mi40), and Michigan 27 (Mi27). Orange points indicate >10-fold differentially expressed contigs (x-axis, as determined via EdgeR); red points indicate significantly differentially expressed contigs (y-axis, FDR-adjusted p < 0.05).

Due to its evolutionary distance from sequenced model organisms, the Diporeia transcriptome remains incompletely annotated. Therefore, DEGs were broadly annotated by putative function using BLASTx via Blast2Go (v.4.0.7). Successfully identified contigs were further assigned to a euKaryotic Orthologous Group, or “KOG” classification (Joint Genome Institute, Walnut Creek, CA, USA). Contigs that received designations of “general function prediction only” (KOG designation “R”) or “function unknown” (KOG designation “S”) were excluded from analysis. Among remaining functionally annotated contigs (n = 696), most were involved in replication, recombination and repair (KOG designation “L”, n = 61), cell wall/membrane/envelope biogenesis (KOG designation “M”, n = 32), or post-translational modification, protein turnover, and chaperones (KOG designation “O”, n = 26, Fig. 4). These three functions were further investigated for potential relevance to viral infection.

Figure 4 Average amphipod transcript expression in relation to LM29173 load.

Average expression (Log10(RPKM+1)) of contigs in transcriptomes associated with above (grey) and below (white) average LM29173 load. Arrows indicate greater (↑) or reduced (↓) average transformed RPKM in transcriptomes with high LM29173 load relative to transcriptomes with low LM29173 load. Contigs are grouped by putative functional annotation (KOG, EuKaryotic Orthologous Groups), and abbreviations correspond to the following functions: (B) chromatin structure and dynamics, (C) energy production and conversion, (D) cell cycle control, cell division, chromosome partitioning, (F) nucleotide transport and metabolism, (I) lipid transport and metabolism, (J) translation, ribosomal structure and biogenesis, (K) transcription, (L) replication, recombination and repair, (M) cell wall/membrane/envelope biogenesis, (N) cell motility, (O) posttranslational modification, protein turnover, chaperones, (T) signal transduction mechanisms, (U) intracellular trafficking, secretion, and vesicular transport, (V) defense mechanisms, (W) extracellular structures.

DEGs involved in replication, recombination, and repair—KOG “L”

The proportion of DEGs involved in modulating DNA synthesis and stability may indicate that CRESS-DNA viruses alter or manipulate cellular replication pathways. This is congruent with the dynamics of circoviral infections in vertebrates, which exploit cellular DNA damage responses through a complex kinase cascade, triggering apoptosis and ultimately facilitating viral replication (Wei et al., 2016). Contigs homologous to unclassified DNA binding proteins, DNA modification enzymes, nucleases, histone structural components, and mobile elements/DNA translocases were differentially expressed. Several DEGs were responsible for chromatin remodeling, indicating a potential correlation between states of nucleosome packaging and viral load. However, many of these transcripts were associated with opposing functions (e.g., DNA methylases and demethylases) and may target different chromatin residues, rendering it difficult to determine if the presence of LM29173 leads to differential transcription.

DEGs involved in cell wall/membrane/envelope biogenesis—KOG “M”

Several homologs of cell-surface receptors and transmembrane transporters including cubilin, calsyntenin, choline transporters, and g-protein coupled receptors were differentially expressed in transcriptomes with high LM29173 load. Contigs putatively involved in carapace biogenesis and the production of other structural/connective tissues (keratin, collagen, and elastin), as well as those involved in cell movement and intracellular transport (actin and myosin) were also significantly differentially expressed. These proteins play central roles in cell growth and replication, and differences in their transcription may be an artifact of natural variability between organisms. However, mis-regulation of these proteins is a well-documented response to many metazoan virus infections (Döhner & Sodeik, 2005; Luftig, 1982; Yan, Zhu & Yang, 2014). For example, cellular entry and trafficking of porcine circoviruses is actin and small GTPase-mediated (Misinzo et al., 2009; Yan, Zhu & Yang, 2014). Myosin is also differentially expressed in subclinical PCV-2 infections and may aid in ATP-dependent intracellular transport of viral particles to the nucleus (Arii et al., 2010; Tomás et al., 2009; Vicente-Manzanares et al., 2009; Xiong et al., 2015).

DEGs involved in post-translational modification, protein turnover, and chaperones—KOG “O”

Intracellular transporters are commonly exploited by vertebrate-associated CRESS-DNA viruses to facilitate entry into the nucleus (Cao et al., 2014; Misinzo et al., 2009). A transcript homologous to Ran (Ras-family related GTP-binding nuclear protein) was overexpressed in transcriptomes with moderate and high LM29173 load, and may be implicated in nucleocytoplasmic transport and regulation of cell cycle progression (Avis & Clarke, 1996; Sazer & Dasso, 2000). Likewise, ubiquitin-conjugating enzyme E2 (UBQ) was overexpressed in libraries with highviral load. This enzyme facilitates covalent attachment of ubiquitin to protein substrates (Liu et al., 2007), and may be exploited by viruses to mis-regulate proteolytic degradation, modify chromatin structure, activate NF-κB and other innate immune mechanisms, or advance G2/M-phase cells into S-phase (Cheng et al., 2014; Gao & Luo, 2006). For example, PCV2 encodes a protein (ORF3) that co-localizes and interacts with E3 ubiquitin ligase, resulting in upregulation of P53 and induction of apoptotic programs, presumably benefiting viral egress (Liu et al., 2007). Knockdown of ubiquitination conjugating enzymes also stalls cells in the G2/M phase, prohibiting PCV2 from accessing S-phase DNA polymerase necessary for viral propagation (Cheng et al., 2014; Liu et al., 2007).

RT-qPCR supports a haplotype-specific relationship between LM29173 load and amphipod gene expression

Viral load correlated with opposite trends in gene expression (average log-transformed RPKM) between Lake Superior and Lake Michigan transcriptomes in all KOG categories with the exception of “chromatin structure and dynamics” (B), “translation, ribosomal structure and biogenesis” (J), and “energy production and conversion” (C; Fig. 4). Unlike Lake Superior libraries, Lake Michigan libraries with high viral load were associated with elevated average RPKM (Fig. 4). Because gene expression in organisms with high viral load may be predicated on population-specific characteristics, we identified 29 common genes differentially expressed in both Lake Michigan stations Mi27 and Mi40 (shared DEGs). However, only one contig (NMHC) was both successfully annotated and potentially affiliated with viral infection (Figs. 5C and 5F). Lake-specific transcriptional profiles confound bulk comparison of gene expression in relation to LM29173 load, and density estimation distributions of individual transcripts indicate that intermediate viral load correlates with increased expression in most KOG classes (Fig. S4). These patterns could indicate that CRESS-DNA virus presence has no appreciable impact on gene expression. Alternatively, because amphipods from Lakes Michigan and Superior belong to potentially phenotypically distinct clades (Pilgrim et al., 2009), these results may indicate that response to environmental and microbial stressors is haplotype-specific.

Figure 5 Relative expression of target genes ACT, UBQ, and NMHC in relation to LM29173 load.

(A–C) Relative expression of target genes β-actin (A; ACT), (B; UBQ) and (C; NMHC) in relation to expression of reference gene elongation factor-1α (EF1A) in specimens from two haplotype clusters (northern and southern) with above (grey) and below (white) average LM29173 copy number (±1 SE). (D–F) Correlation between viral load and relative expression of ACT (D), UBQ (E), and NMHC (F) in relation to EF1A reference gene expression. Quantities of target amplicons were standardized by reference gene EF1A using the following equation: (TargetRT–TargetNRT)/(EF1ART − EF1ANRT), where RT and NRT indicate samples that have been reverse transcribed via Superscript III (Invitrogen, Carlsbad, CA, USA), or not reverse transcribed (no-RT control), respectively. (G) Gene expression (reads per kilobase of transcript per million mapped reads; RPKM) of ACT, NMHC, and UBQ per transcriptome library in each of three stations: Lake Superior station 066 (SU066), Lake Michigan station 40 (Mi40) and Lake Michigan station 27 (Mi27). Libraries are ranked from left to right by increasing LM29173 load.

RT-qPCR quantification of ACT, NMHC, and UBQ confirmed opposite trends in contig expression in correlation with above average viral load among amphipods from Lake Michigan and Superior (Fig. 5). Contigs DN12114c1g3i8 (230 nt), DN12352c3g10i1 (1,229 nt), and DN135c0g1i1 (309 nt) exhibited sequence similarity to ACT from penaeid blue shrimp (Litopenaeus stylirostris; BLASTx, e-value 6 × 10−50), NMHC from freshwater amphipods (Hyalella azteca; e-value 3.0 × 10−8), and UBQ from freshwater amphipods (H. azteca; e-value 4 × 10−52), respectively. Relative expression of ACT, NMHC, and UBQ did not significantly correlate with viral load, suggesting that LM29173 does not likely specifically alter transcription of these genes, or the practice of whole-organism RNA extraction obscures cell-specific response(s) to viral presence (Fig. 5). However, relative expression of target genes varied in relation to amphipod population. Organisms associated with the southern haplotype clade exhibited greater average NMHC and UBQ expression, but diminished average ACT expression in concurrence with high viral copy number, relative to amphipods associated with the northern haplotype clade (p > 0.05, Welch’s t-test for all pairwise comparisons).

Expression of target genes ACT, NMHC, and UBQ was standardized to expression of contig DN11198c0g1i1 (723 nt), a homolog of elongation factor 1 − α (EF1A) from H. azteca (BLASTx, e-value 2 × 10−139). This constitutively expressed gene has been validated as an invariant internal RT-qPCR control under experimental conditions in decapods (Leelatanawit et al., 2012), and provided adequate reference to the baseline transcriptional activity of Diporeia, as expression did not correlate with amphipod wet weight, lake, or LM29173 load (Fig. S5). Variability in ACT, NMHC, and UBQ expression may be a result of nonspecific RNA extraction, which confounds assessments of specific impacts(s) of viral presence on single tissue types or cells. Additionally, RT-qPCR cannot detect changes in the intracellular localization of myosin subunits nor the state of polymerization of actin subunits, and additional investigation via microscopy and proteomics may be warranted.

Expression of amphipod innate immunity regulators and effectors

Diporeia transcriptomes were surveyed for homologs of genes involved in crustacean innate immunity to determine if LM29173 presence correlates with immune-specific gene expression. About 148 homologs (BLASTx e < 1 × 10−5) were identified and exhibited >2-fold differential expression in both Lake Michigan station Mi27 and Mi40. Genes involved in stress response (heat shock or oxidative stress response), immune-specific signaling and post-translational modification, and immune-associated cell structure, mobility, and intracellular trafficking mechanisms were consistently overexpressed in Lake Michigan libraries with high viral load (Fig. S6). Correlative evidence that these immune-related genes are overexpressed in association with high viral load does not preclude the possibility of other co-occurring immune demands. Therefore, is unclear if overexpression of these genes are a product of environmental stress, or if they are specific responses to viral infection.

It remains unclear to what extent LM29173 impacts lake-wide Diporeia population dynamics. However, the presence of LM29173 among stable amphipod populations and negligible changes in expression of specific amphipod disease pathways in relation to this viral genotype likely indicate that LM29173 is not solely responsible for Diporeia decline in the Laurentian Great Lakes. We stipulate that CRESS-DNA viruses associated with Diporeia may play a subtle role in altering amphipod physiology, if any. This observation corroborates data from well-characterized mammalian CRESS-DNA viruses (PCV1; Allan & Ellis, 2000; TTV; Okamoto, 2009), which often manifest asymptomatically in healthy host tissue. We speculate that LM29173, like other CRESS-DNA viruses, may evade host clearance, attenuate innate immune responses, or elicit host tolerance through post-transcriptional or translational gene regulation, ultimately establishing persistent and asymptompatic infections (Brajão de Oliveira, 2015; Okamoto, 2009). This hypothesis may explain the universal prevalence and diversity of these viruses in aquatic ecosystems, as observed by metaviromic sequencing.

In summary, while LM29173 load does not correlate with significant differential expression of specific gene pathways, transcriptional changes in genes involved in several physiological functions, including innate immunity, are detectable and specific to distinct haplotype clades. To our knowledge, this study communicates the first investigation of the transcriptional relationship between invertebrates and associated CRESS-DNA viruses in natural ecosystems. This study also provides several potential transcriptional targets for further investigation of gene/pathway-specific inquiries to determine if the bulk of these novel viruses have little effect on metazoan gene expression or physiology.

Supplemental Information

Supplemental Information 1 Quantitation of LM29173 prevalence and load.

(A–C) Load of LM29173 copy number animal−1 or mg−1 (± 1SE between quadruple technical replicates) in organisms from stations Superior 066 (A, Su066), Michigan 40 (B, Mi40), and Michigan 27 (C, Mi27). (D) Boxplot illustrating distribution of viral load among amphipods from three stations. Outliers (< average and > average LM29173 genome copies animal−1) were selected for transcriptome preparation.

Click here for additional data file.

Supplemental Information 2 Ranked abundance of reads (log10 transformed) mapped to individual contigs.

Click here for additional data file.

Supplemental Information 3 Correlation matrix (multi-dimensional scaling; MDS) exhibiting the normalized degree of variation between libraries grouped by lake (A) and station (B).

Plots were generated via CLC workbench (v. 8.5.1, Qiagen, Hilden, Germany) with default parameters.

Click here for additional data file.

Supplemental Information 4 Density estimation distributions of contigs in relation to LM29173 load.

(A) Density estimation distributions depicting expression (Log10(RPKM + 1)) of contigs associated with 15 KOG classes at a range of viral loads (LM29173 load organism−1; n = 12 transcriptomes). (B) Density estimation distributions of RT-qPCR target contigs UBQ and ACT depicting expression (Log10(RPKM + 1)) at a range of viral loads (LM29173 load organism−1; n = 12 transcriptomes).

Click here for additional data file.

Supplemental Information 5 Expression of reference gene elongation factor-1α (EF1A) relative to organism wet weight and LM29173 load mg−1 (Log10 + 1 transformed).

EF1A expression is reported as the quantitated difference between reverse transcribed and non-reverse transcribed RNA extractions. i.e. EF1ART–EF1ANRT, where RT and NRT indicate samples that have been reverse transcribed via Superscript III (Invitrogen, Carlsbad, CA, USA) or not reverse transcribed (no-RT control), respectively.

Click here for additional data file.

Supplemental Information 6 Heat map depicting expression (RPKM) of differentially expressed contigs (>2 fold over or underexpressed in both Lake Michigan stations) homologous to invertebrate innate immunity effectors (BLASTx e-value < 1 × 10−5) standardized by total RPKM.

Contigs are grouped by putative function: (A) immune response regulation, (B) antimicrobial peptides, (C) phagocytosis, (D) protease/chitinase, (E) JAK/STAT pathway, JNK pathway, (F) toll pathway, (G) prophenoloxidase system, (H) stress/oxidative damage response, (I) transcription factors, (J) signaling, post-translational modification, (K) cell structure, mobility, and intracellular trafficking, (L) receptors, and (M) cell cycle. Image generated by web-based visualization tool, Morpheous (Broad Institute, Cambridge, MA, USA).

Click here for additional data file.

Supplemental Information 7 Amphipod collection site details (congruent with EPA-Great Lakes National Program Office designated stations).

Organisms were acquired via Ponar benthic sampler from the R/V Lake Guardian between August–September, 2014 (n = 98). RT-qPCR (n) refers to the number of samples per station allocated to RT-qPCR. HTS (high throughput sequencing) refers to stations where amphipods were collected for transcriptome preparation and sequencing. Haplotype was determined via cytochrome c oxidase I (COI) sequencing (Pilgrim et al., 2009).

Click here for additional data file.

Supplemental Information 8 Transcriptome assembly statistics.

Transcriptome assembly statistics. Read libraries were pooled and assembled de novo using de Bruijn graphs integrated into Trinity v.2.4.0, a tripartite assembly program (software modules: Inchworm, Chrysalis, and Butterfly) implemented on the Galaxy bioinformatics platform per default parameters (National Center for Genome Analysis Support, Indiana University Pervasive Technology Institute; Trinity –max_memory 240G –CPU 8 –normalize_reads –monitoring –seqType seq_type –single singlefile or –left left_file –right right_file).

Click here for additional data file.

Supplemental Information 9 NCBI protein database keyword queries.

Reference sequences fitting both keyword 1 and 2 (e.g. “Crustacea” + “Toll like receptor”) were collected and collated from the NCBI protein repository as a BLAST database to identify transcriptome contigs affiliated with putative immune functions.

Click here for additional data file.

Supplemental Information 10 qPCR and RT-qPCR primer/probe sequences and reaction parameters.

All reactions were 25ul and included SsoAdvanced™ Universal Probes Supermix (Bio-Rad Laboratories, Hercules, CA, USA) with 2 μM primer/probe oligo (Eurofins Scientific, Luxembourg City, Luxembourg) per reaction. Reaction efficiencies of duplex reactions were comparable to those when reactions were run independently. Quantities of target amplicons were standardized by reference gene EF1A using the following equation: (TargetRT–TargetNRT)/(EF1ART–EF1ANRT), where RT and NRT indicate samples that have been reverse transcribed via Superscript III (Invitrogen, Carlsbad, CA, USA), or not reverse transcribed (no-RT control), respectively. LLOD specifies average lower limit of detection (Ct) across all runs containing the indicated primer/probe set and the corresponding amplicon copy number. Samples with Ct values > LLOD were designated “no detection” (negative). Average threshold Ct indicates ΔRn where quantity was determined (per StepOnePlus software v. 2.3; Foster City, CA, USA).

Click here for additional data file.

Supplemental Information 11 Summary of Diporeia transcriptomes.

Reads were trimmed using CLC workbench (v. 8.5.1, Qiagen, Hilden, Germany: quality limit 0.05, no ambiguous nucleotides, maximum read length 251 nt, discard reads <50 nt), and assembled de novo using Trinity on the Galaxy bioinformatics platform per default parameters (National Center for Genome Analysis Support, Indiana University Pervasive Technology Institute, USA).

Click here for additional data file.

Supplemental Information 12 Raw quantitation data (Ct values) for transcript-specific RT-qPCR.

Click here for additional data file.

We would like to thank Dr. Gary Blissard and Elliot Jackson for indispensable insight into transcriptome analysis and manuscript revision, and Dr. Jim Watkins and the crew of the R/V Guardian (2014) for assistance with sample collection and processing. We also worked closely with the Core Genomics Facility (BRC) at Cornell for sequencing support.

Additional Information and Declarations

Competing Interests

Author Contributions

DNA Deposition

Data Availability

The authors declare that they have no competing interests.

Kalia S.I. Bistolas conceived and designed the experiments, performed the experiments, analyzed the data, wrote the paper, and prepared figures and/or tables.

Lars G. Rudstam conceived and designed the experiments, contributed reagents/materials/analysis tools, and reviewed drafts of the paper.

Ian Hewson conceived and designed the experiments, analyzed the data, contributed reagents/materials/analysis tools, and reviewed drafts of the paper.

The following information was supplied regarding the deposition of DNA sequences:

Transcriptome sequence data were deposited in Genbank (https://www.ncbi.nlm.nih.gov/genbank/) under accession numbers: PRJNA379017; SRR5341776–SRR5341788.

The following information was supplied regarding data availability:

The raw quantitation data (Ct values) for transcript-specific RT-qPCR has been uploaded as Supplemental Dataset Files.

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
