# Peer review of "Gene expression of benthic amphipods (genus: Diporeia) in relation to a circular ssDNA virus across two Laurentian Great Lakes"

_PeerJ, doi:10.7717/peerj.3810_

## Round 0.1 · original submission · Minor Revisions

I really enjoyed reading this manuscript and found that it was well written and elucidated. Overall I feel there are a few areas that paper can be improved. In particular I agree with reviewer 1 that you need to be more transparent about your assembly. I agree the assembly may be a bit fragmented and this needs to be reported.

Reviewer 1 ·

Basic reporting

no comment

Experimental design

This study reports the transcriptome analysis of environmentally sampled Diporeia exhibiting differences in CRESS-DNA viral load. This report appears to be the first to explore transcriptional responses to infection with this virus in invertebrates. An analysis of differential expression was performed using RNA-Seq and standard methods, supported by qPCR data across 3 selected genes. DEGs between animals with high and low viral load are discussed in the context of their KOG classification and association with immune response. Interestingly, transcriptional changes in genes such as those involved in innate immunity were observed among the 3 sampling sites investigated.

This is an interesting article within the scope of this journal that is well written and clearly presented. The authors provide a good amount of data to support their findings and I have been able to confirm the raw sequence submissions at NCBI.

The main considerations relate to information surrounding the analysis of the RNA-Seq data in this study. Some detailed comments are listed below.

Results and discussion
It appears that there are multiple extant Diporeia species, can species be differentiated using COI gene analysis? It is unclear in the manuscript if the Diporeia sampled in the study are of the same species, though this was assumed with the mention of sub-species in line 189. The authors mention in Line 205 “the lack of clustering between samples is likely as a result of high variability between organisms”, this might be expected of the comparisons if the samples are distantly related. I feel that the term “haplotype clades” needs to be clarified in the context of defining relatedness between samples.

Line 200: The reported contig N50 is rather low (290 nt), and read mapping results are quite poor (as low as 16% and not greater than 62%) as outlined in Table S4. It is possible that the reference assembly is still rather fragmented. The authors should clarify how this has been handled in the analysis (assembly metrics file seems to be missing – see General comments).

Sample MI40(130) has a very low sequence yield compared to the other samples which may have introduced some bias to the DE analysis, unduly causing some dissimilarity with genes expressed in low levels. The authors should provide some support for retaining this sample in the study.

Line 108: The authors noted that reads mapping to the SILVA database were excluded from the assembly. Were there reads mapping in high abundance to database entries for potential pathogens of Diporeia, that may suggest a possible co-infection? If so, this may be interesting to mention in the manuscript.

It appears from Table S4 that a large proportion of reads map to the rRNA database, which seems to impact heavily on the overall sequencing depth. I would suggest that the authors offer a comparative example with another similar study in terms of their sequence yield.

Validity of the findings

no comment

Additional comments

Files and Figures
Summary of transcriptome assembly and annotation statistics doesn’t appear to be included in the Supplementary Table S4 file as the file name suggests. This file seems to contain only information on raw read and trimming/mapping results. These results should be included.

Supplementary Fig 2 appears to be missing? Can’t seem to locate the transcriptome stats mentioned in the manuscript.

Supplementary file Raw_Ct_peerJ.d0.xlsx it appears should be named Supplementary Table 3.

Minor grammatical correction required with the use of hyphenation being inconsistent for “over- and under-expressed”.

·

Basic reporting

The MS is well written and easy to follow. The figures and tables are well laid out and i don not see any issues with this.

Experimental design

The experimental design is good.

Validity of the findings

Data is is statistically sound based on the methods used by the authors.

Additional comments

I enjoying reading this MS. It is well laid out and relatively easy to follow. I like the concept of the MS and the aim to address the poor knowledge of the viral dynamics of LM29173 in Diporeia sp.

I have a few minor suggestions and questions

1) Abstract and else where: 0-3x10E6. 0x10E6 = 0. So perhaps change to "up to 3x10E6"
2) Lines 135-139: can this be put in a supplementary table to make it easy to follow?
3) Line 330-331: change "(PCV1; Allan & Ellis 2000); TTV; Okamoto 2009)" to "(PCV1; Allan & Ellis 2000; TTV; Okamoto 2009)"
4) In your transcriptome analsysis did you find other CRESS DNA viral transcripts or RNA viral transcripts that may be co-infecting Diporeia sp and thus some of the up or down regulation may be as a result of a complex of viruses rather an only LM29173?

---

## Round 0.2 · accepted · Accept

You have answered all queries very well and I am happy for this manuscript to proceed to publication.